# Active Pharmacovigilance Study: A Follow-Up Model of Oral Anti-Cancer Drugs under Additional Monitoring

**Sofia Pinto Carvalho da Silva** [1,†], **Mafalda Jesus** [2,3,†], **Fátima Roque** [3,4,5], **Maria Teresa Herdeiro** [6], **Rita Costa e Sousa** [7], **Ana Paula Duarte** [2,3,8] **and Manuel Morgado** [2,3,4,5,9,*]

1.   Pharmaceutical Services of Local Healthcare Unit of Matosinhos, 4464-513 Senhora da Hora, Portugal
2.   Health Sciences Faculty, University of Beira Interior (FCS-UBI), 6200-506 Covilhã, Portugal
3.   Health Sciences Research Centre, University of Beira Interior (CICS-UBI), 6200-506 Covilhã, Portugal
4.   School of Health Sciences, Polytechnic Institute of Guarda, 6300-749 Guarda, Portugal
5.   Research Unit for Interior Development, Polytechnic of Guarda (UDI-IPG), 6300-749 Guarda, Portugal
6.   Institute of Biomedicine, University of Aveiro (iBIMED-UA), 3810-193 Aveiro, Portugal
7.   Hematology Service, University Hospital Center of Coimbra, 3004-561 Coimbra, Portugal
8.   UFBI—Pharmacovigilance Unit of Beira Interior, University of Beira Interior, 6200-506 Covilhã, Portugal
9.   Pharmaceutical Services of University Hospital Center of Cova da Beira, 6200-251 Covilhã, Portugal
*    Correspondence: mmorgado@fcsaude.ubi.pt
†    These authors contributed equally to this work.

**Abstract:** Adverse drug reactions (ADRs) are responsible for almost 5% of hospital admissions, making it necessary to implement different pharmacovigilance strategies. The additional monitoring (AM) concept has been highlighted and intended to increase the number of suspected ADRs reported, namely in medicines with limited safety data. A prospective, descriptive study of active pharmacovigilance (AP) was conducted between 2019 and 2021 in the Local Health Unit of Matosinhos (LHUM) (Porto, Portugal). A model of AP for medicines under AM, namely oral antineoplastic agents, was designed. Follow-up consultations were performed, and adverse events (AEs) data were collected. The overall response to the treatment was evaluated through the Response Evaluation Criteria in Solid Tumors (RECIST) 1.1 criteria. A total of 52 patients were included in the study, and 14 antineoplastic drugs under AM were analyzed. Of the total number of patients included, only 29 developed at least one type of toxicity. Hematological disorders were the most reported suspected ADR. However, only four patients interrupted their treatment due to toxicity. After 12 months of treatment, most patients had disease progression, which was the main reason for therapy discontinuation. This AP model played an important role in the early detection of AEs and, consequently, contributed to better management of them. Increasing the number of suspected ADR reports is crucial for drugs with limited safety data.

**Keywords:** pharmaceutical intervention; adverse drug reactions; risk management; black inverted triangle; patient safety and oral chemotherapy

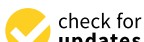



## 1. Introduction

Adverse drug reactions (ADRs) are the fifth leading cause of death in hospital settings [1]. According to data from the European Union (EU), ADRs are responsible for almost 5% of hospital admissions [1]. In 2005, results showed that ADRs were responsible for 197,000 deaths annually [2]. Moreover, the impact and management of ADRs also constitute significant economic burdens [3–5]. ADRs are almost always associated with antineoplastic therapy and are widely accepted as unavoidable by both patients and healthcare providers. In several studies, all patients (100%) receiving anticancer drugs had at least one ADR, and a normal range of 2–7 ADRs per cancer patient was referred [6–8]. Currently, in daily oncology clinical practice, many side effects can be attributed to antineoplastic targeted drugs. Although different from the well-known side effects of chemotherapy,

these adverse effects can seriously compromise patients' quality of life and may even lead patients to discontinue or request a change in treatment. It is important to study the toxicity of antineoplastic targeted therapies because most of the ADRs included in the summary of product characteristics (SmPC) come from pivotal clinical trials: the list of ADRs identified in these trials is unlikely to be comprehensive because, in clinical trials, drugs are tested under controlled conditions in selected patients [9].

Several authorities, such as the European Medicines Agency (EMA), the Member States of the EU, and the European Commission have contributed to the development of pharmacovigilance activities, such as the introduction of Regulation (EU) No 520/2012 of 19 June 2012 [10]. In this context, among other measures implemented through this legislation, the additional monitoring (AM) concept is highlighted [1,10]. This concept intends to increase the number of suspected ADR notifications reported, namely in medicines that have limited safety data, thus requiring closer monitoring by regulatory authorities. Despite additional safety monitoring, these medicines are considered safe since only medicines with benefits that outweigh the risks are introduced into the European market [11]. AM status includes medicines in the following cases: (i) medicines that contain a new active substance authorized in the EU after 1 January 2011; (ii) biological medicines, such as vaccines or plasma-derived medicines, authorized in the UE after 1 January 2011; (iii) medicines with a conditional approval (cases in which the company that markets the medicine must provide more data about it) or medicines authorized under exceptional circumstances (when there are specific reasons why the company cannot provide a comprehensive set of data); (iv) medicines for which further studies are needed (e.g., to provide more data on the long-term use of the medicine or a rare side effect seen during clinical trials); (v) medicines authorized with specific obligations on the recording of suspected ADRs. Other medicines can be included in the list of medicines under AM based on advice from the EMA's Pharmacovigilance Risk Assessment Committee (PRAC) [11]. The PRAC is the European Medicines Agency committee responsible for assessing and monitoring the safety of human medicines. This committee is responsible for reviewing the list of medicines under AM every month. Medicines can be included on the list when approved for the first time or at any time during their life cycle and remain on it for 5 years or until being removed under the decision of the PRAC [5]. Medicines under AM are labeled with a black inverted triangle (▼) in their SmPC and package leaflet, accompanied by a brief explanatory statement, which allows for the quick identification of these medicines by patients and healthcare professionals [12,13].

Since a substantial part of these medicines is provided by hospital pharmacists, in an outpatient regime, the role of these healthcare professionals is extremely relevant along with the collaboration of patients and other healthcare professionals, such as doctors and nurses [14]. In the literature, there is an evident lack of studies about the real-world safety, efficacy/effectiveness, and cooperation between healthcare professionals in the management of the ADRs of medicines under AM. Although these medicines are already on the market and almost all their adverse effects, identified by clinical trials, are already described in the respective SmPC, the fact that they have the status of drugs under AM justifies the realization of real-world pharmacovigilance studies. Real-world studies are performed in real clinical practice settings and are better able to assess the efficacy/effectiveness and safety of drugs when used in real life by patients and physicians.

The under-reporting of ADRs in oncology can be addressed through proactive forms of pharmacovigilance and multidisciplinary collaborations [9]. This study intends to develop, implement, and evaluate the impact of an active pharmacovigilance (AP) model for oral antineoplastic agents labeled with the black inverted triangle, through an internal procedure involving the pharmaceutical services, the oncology service, and the clinical hematology service. Using a multidisciplinary approach, we want to improve the early identification of adverse events (AEs) associated with these medicines and assess these AEs. We also intend to educate patients or caregivers on the importance of being aware of the signs/symptoms of possible AEs that occur during treatment and to encourage their

reporting. By improving knowledge of the AEs of these drugs, better risk management is expected.

## 2. Materials and Methods

### 2.1. Study Design

A prospective, descriptive study of AP was conducted between 2019 and 2021 in the Local Health Unit of Matosinhos (LHUM) (Porto, Portugal). The study was supported by the collaboration of 3 services, namely: the pharmaceutical services that integrate the hospital pharmacy, the oncology service, and the clinical hematology service. This study was approved by the ethics committee and authorized by the Board of Directors of the Hospital Unit (NS 64/CE/JAS-07/06/2019). All subjects provided written informed consent for participation. The eligible patients were those 18 years or older whose therapy included at least one oral anti-cancer drug submitted to AM. The active substances of medicines under AM considered in the study were alectinib, cabozantinib, entrectinib, ixazomib, lenalidomide, lorlatinib, niraparib, osimertinib, palbociclib, ribociclib, trametinib in association with dabrafenib, vandetanib, venetoclax, and trifluridine plus tipiracil.

### 2.2. Risk Minimization Measures

The initial phase of this study was performed through the implementation of risk minimization measures for AEs caused by the selected medicines. Risk minimization emerged from the need to inform patients and raise awareness among healthcare professionals [15]. According to the guidelines on good pharmacovigilance practices, the goal of risk minimization measures is to facilitate informed decision-making to support risk minimization when prescribing, dispensing, and/or using a medicinal product [16]. The successful implementation of risk minimization measures guarantees the principles of risk management where the benefits of a medicinal product exceed the risks by a large margin [17].

In this context, a list of the medicines under AM prescribed in the hospital was created. This list is periodically updated and published together with a poster that contains information about the medicines under the AM concept and about the meaning of the black inverted triangle, and it is shown in a place visible to patients and healthcare professionals. In the hospital pharmacy, these medicines are identified with an orange label in their storage facility. An information leaflet that contained several details about these identified medicines—mainly the most frequent AEs reported in their SmPC, recommendations for prescribing, and preventive measures as well as signs/symptoms to be monitored—was also written for healthcare professionals.

### 2.3. Adverse Events Monitoring

After the prescription of the drugs under AM, the patients had their first pharmaceutical consultation with a hospital pharmacist. Aspects related to the dosage, drug–drug and drug–food interactions, storage conditions, and most relevant excipients (e.g., lactose and sodium), which could potentially result in some interactions, were explained. In this first consultation, the pharmacist also clarified the concept of medicines under AM and the main purpose of this study. The signs and symptoms of the AEs and recommendations for their management were also highlighted to the patients. In the following visits to the hospital, after their report in medical and pharmaceutical follow-up consultations, the AEs were monitored by the hospital pharmacists. More details are discussed below. The next follow-up consultations were carried out according to the evaluation cycle of each drug. For instance, palbociclib and ribociclib are evaluated every 15 days during the first two cycles. If no toxicity is detected, the frequency of the evaluation becomes monthly.

The 4 minimum criteria to report an adverse drug reaction were: an identifiable patient, a suspected medicine, a suspected reaction, and an identifiable reporter. According to the guideline on good pharmacovigilance practices (GVP) published by the European Medicines Agency and the Heads of Medicines Agencies, an adverse reaction is character-

ized by the fact that a causal relationship between a medicinal product and an occurrence is suspected [18]. For regulatory reporting purposes, if an event is spontaneously reported—even if the relationship is unknown or unstated—it meets the definition of an adverse reaction. Therefore, all spontaneous reports submitted by healthcare professionals or consumers are considered suspected adverse reactions, since they convey the suspicions of the primary sources, unless the reporters specifically state that they believe the events to be unrelated or that a causal relationship can be excluded [18].

In this active pharmacovigilance study, alongside the spontaneous reports of ADRs made by health professionals and patients (passive pharmacovigilance), data on ADRs obtained using the active pharmacovigilance model developed for this purpose were predominantly collected. In this model of active pharmacovigilance, healthcare professionals themselves (physicians and pharmacists, in our study) deliberately question the patients and other healthcare professionals (e.g., other physicians and nurses) about the occurrence of ADRs and obligatorily register them in a database created for this purpose. The use of active pharmacovigilance models to identify ADRs has been highlighted in pharmacovigilance activities, as they are able to detect a greater number of patients (and with more diverse characteristics) with ADRs and, consequently, reduce the chronic underreporting of adverse drug events [19].

The collection of AEs—and, consequently, their monitoring—was supported by several tools developed by the hospital pharmacists in collaboration with the oncology service, namely:

An AE monitoring card was a card delivered to the patient during the first pharmaceutical consultation. This card contained the medicine's name, highlighting the fact that the patient was taking a medicine under AM and the most frequent AEs requiring monitoring. This card allowed the patient to register several points, mainly: the exact start and the end of the AE occurrence, whether the physician was contacted by the patient during the occurrence, and what action was taken after the contact. In addition, the card included telephone numbers, such as the hospital unit, pharmaceutical services, oncology and clinical hematology services numbers, and other emergency contacts. The electronic addresses of INFARMED and Porto's Pharmacovigilance Unit were also mentioned to encourage an increase in suspected ADR reports by patients.

The patient information leaflet (PIL) was a supporting document that explained, in more detail, the information mentioned in the AE monitoring card related to medicine under AM.

A webpage discussing the active pharmacovigilance of medicines under AM was also available on the hospital intranet. A page on the LHUM intranet was created with all the information and documents related to the study (informed consent form, PIL, AE monitoring card, and SmPC of the medicines included). It was accessible to all hospital pharmacists. It also included a simplified database of the AEs reported during the study period and direct access to the platforms used for reporting suspected ADRs, such as Porto's Pharmacovigilance Unit.

The Medicines under AM Monitoring platform was a platform restricted to the study investigators that contained all data collected from AE monitoring, namely: dose reductions, the temporary or permanent suspension of treatment, and the causes that originated it. This record was compiled by drug and by patient. Other data, such as the information reported by the patient in medical and pharmaceutical follow-up consultations (even over the telephone) and AE monitoring card records, were mentioned. These results were used for the next phase.

### 2.4. Effectiveness and Safety Monitoring Database

The data obtained on the safety and effectiveness of drugs was discussed among healthcare professionals following the study. After data analysis, an effectiveness and safety monitoring database was created, allowing for the generation of clinical outcomes experienced by the patients. The Common Terminology Criteria for Adverse Events (CTCAE, also

called common toxicity criteria (CTC)) was used to classify the AE of antineoplastic drugs according to the severity grades (1 = mild, 2 = moderate, 3 = severe, 4 = life-threatening, and 5 = death) [20]. In the CTCAE, an adverse event is defined as any abnormal clinical finding temporally associated with the use of a therapy for cancer; causality is not required. These criteria are used for the management of chemotherapy administration and dosing and to provide standardization and consistency in the definition of treatment-related toxicity. In addition, the overall response to the treatment was evaluated through the Response Evaluation Criteria in Solid Tumors (RECIST) 1.1 criteria [21].

A scheme of the methods section is displayed in Figure 1.

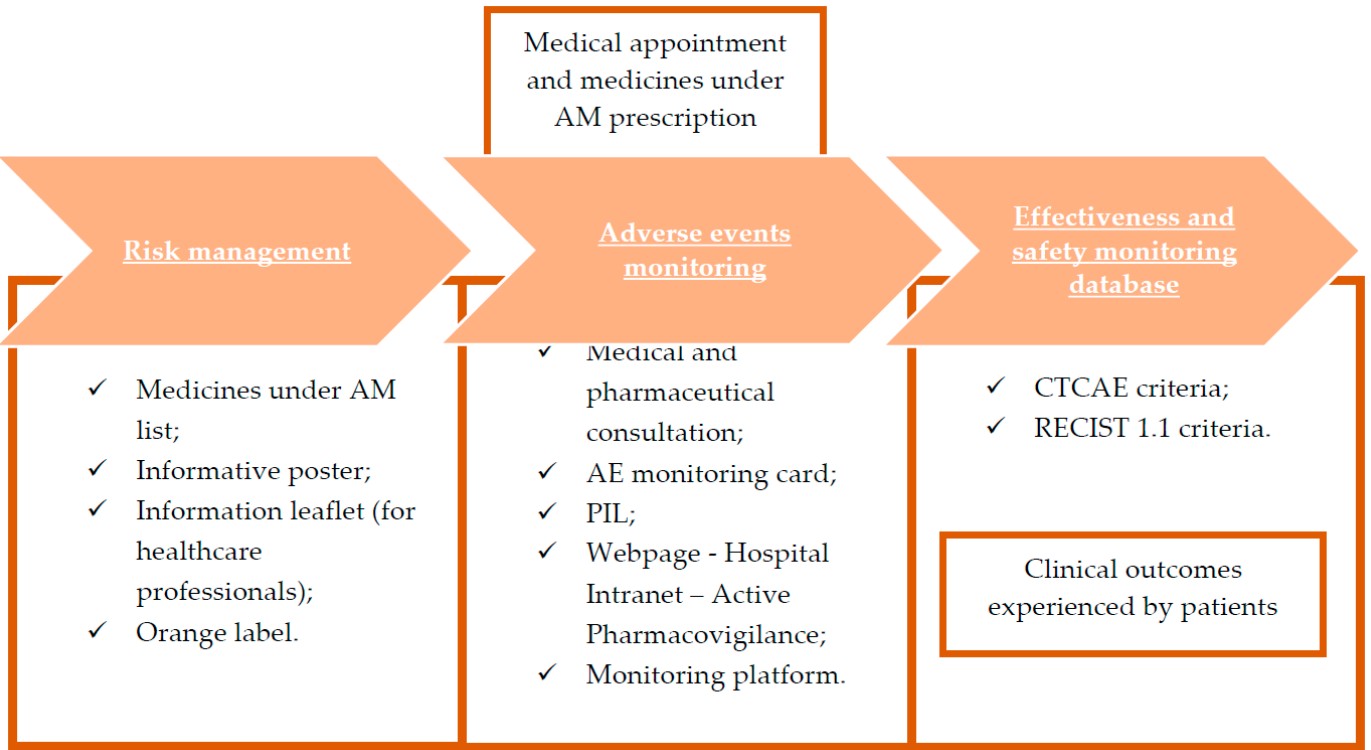

**Figure 1.** Methodology representation. AE, adverse events; AM, additional monitoring; CTCAE, Common Terminology Criteria for Adverse Events; PIL, patient information leaflet; RECIST, Response Evaluation Criteria in Solid Tumors.

### 3. Results

A total of 52 patients were included, and 14 drugs under AM were studied according to their base pathology. Table 1 describes the number of patients involved according to their diagnosis and the medicine under AM that was prescribed. Except for lenalidomide and trifluridine plus tipiracil, all other oral anticancer drugs are classified as targeted drugs consisting of small molecule inhibitors (e.g., tyrosine kinase inhibitors). Of the total number of drugs under AM analyzed, trifluridine plus tipiracil was the most prescribed drug. A total of 9 patients were monitored in terms of AEs in the treatment of colon adenocarcinoma, rectal adenocarcinoma, and gastric adenocarcinoma. Osimertinib and venetoclax were prescribed to 7 patients each for the treatment of lung adenocarcinoma and chronic lymphocytic leukemia, respectively. Cabozantinib, entrectinib, and trametinib, in association with dabrafenib and vandetanib, were prescribed to only one patient.

Of the 52 patients, 31 were female and 21 were male. Palbociclib and ribociclib were prescribed to female patients, considering their base pathology. Other drugs, such as cabozantinib, lorlatinib, and niraparib, were also prescribed to females only, while entrectinib, vandetanib, and the association trametinib plus dabrafenib were prescribed to

males only. The predominant age groups were 50–60 and 60–70 years, with 15 patients in each group. The results are presented in Tables 2 and 3.

**Table 1.** Medicines under AM included in the study according to their base pathology.

| Medicines under AM | Base Pathology | Number of Patients |
|---|---|---|
| Alectinib | Lung adenocarcinoma | 4 |
| | Lung squamous cell carcinoma | 2 |
| Cabozantinib | Hepatocellular carcinoma | 1 |
| Entrectinib | Lung adenocarcinoma | 1 |
| Ixazomib | Multiple myeloma | 3 |
| Lenalidomide | Multiple myeloma | 5 |
| Lorlatinib | Lung adenocarcinoma | 2 |
| Niraparib | High-grade serous ovarian carcinoma | 2 |
| Osimertinib | Lung adenocarcinoma | 7 |
| Palbociclib | Breast cancer | 3 |
| Ribociclib | Breast cancer | 4 |
| Trametinib + Dabrafenib | Lung adenocarcinoma | 1 |
| Vandetanib | Medullary thyroid carcinoma | 1 |
| Venetoclax | Chronic lymphocytic leukemia | 7 |
| Trifluridine + Tipiracil | Colon adenocarcinoma | 5 |
| | Rectal adenocarninoma | 2 |
| | Gastric adenocarcinoma | 2 |
| N Total | | 52 |

**Table 2.** Patients categorized by gender and the medicine under AM that was taken.

| Number of Patients | Gender | | |
|---|---|---|---|
| Medicines under AM | F | M | N Total |
| Alectinib | 3 | 3 | 6 |
| Cabozantinib | 1 | | 1 |
| Entrectinib | | 1 | 1 |
| Ixazomib | 2 | 1 | 3 |
| Lenalidomide | 4 | 1 | 5 |
| Lorlatinib | 2 | | 2 |
| Niraparib | 2 | | 2 |
| Osimertinib | 5 | 2 | 7 |
| Palbociclib | 3 | | 3 |
| Ribociclib | 4 | | 4 |
| Trametinib + Dabrafenib | | 1 | 1 |
| Vandetanib | | 1 | 1 |
| Venetoclax | 3 | 4 | 7 |
| Trifluridine + Tipiracil | 2 | 7 | 9 |
| N total | 31 | 21 | 52 |

F, female; M, male.

**Table 3.** Patients categorized by gender according to their age group.

| Number of Patients | Gender | | |
|---|---|---|---|
| **Age Group (Years)** | **F** | **M** | **N Total** |
| 40–50 | 2 | | 2 |
| 50–60 | 9 | 3 | 12 |
| 60–70 | 9 | 6 | 15 |
| 70–80 | 8 | 7 | 15 |
| 80–90 | 3 | 5 | 8 |
| N total | 31 | 21 | 52 |

F, female; M, male.

A total of 29 (55.8%) patients developed at least one type of suspected toxicity to the drugs prescribed. In patients who reported only one toxicity, most patients developed grade 2 toxicity. Overall, 5 patients developed grade 1 toxicity, 15 patients developed grade 2 toxicity, and 9 patients developed grade 3 toxicity. Grade 3 toxicity was mostly related to the development of hematological disorders in patients taking ixazomib, palbociclib, or ribociclib. Patients taking alectinib, lenalidomide, osimertinib, or trifluridine plus tipiracil developed more than one type of toxicity. Indeed, gastrointestinal disorders are common in patients taking alectinib and trifluridine plus tipiracil [22,23], as also reported in our study. Only four patients had more than one toxicity. For instance, the patient that was taking cabozantinib developed mucositis grade 3 toxicity and hypertension grade 4 toxicity. The other three patients within whom more than one type of toxicity was reported were taking alectinib, entrectinib, and vandetanib. Table 4 describes the type of toxicity associated with the medicines under study as well as their grade (for additional information about the molecular structure of the studied oral antineoplastic drugs, see Supplementary Material).

**Table 4.** Type and grade of suspected toxicity associated with the medicines in this study.

| Medicines under AM | Type of the 1st Suspected Toxicity | Grade of the 1st Suspected Toxicity | Type of the 2nd Suspected Toxicity | Grade of the 2nd Suspected Toxicity | Number of Patients |
|---|---|---|---|---|---|
| Alectinib | Gastrointestinal disorders | 2 | | | 1 |
| | Liver disorders | 2 | | | 1 |
| | Lung disorders | 2 | | | 1 |
| | Kidney disorders | 2 | Liver disorders | 3 | 1 |
| Cabozantinib | Mucositis | 3 | Hypertension | 4 | 1 |
| Entrectinib | Hypotension | 2 | Vomiting | 2 | 1 |
| Ixazomib | Hematological disorders | 3 | | | 1 |
| Lenalidomide | Anorexia | 2 | | | 1 |
| | Hematological disorders | 1 | | | 1 |
| | Neurological disorders | 2 | | | 1 |
| Lorlatinib | Dyslipidemia | 2 | | | 1 |
| Osimertinib | Mucositis | 3 | | | 1 |
| | Paronychia | 2 | | | 1 |
| | | 3 | | | 1 |
| Palbociclib | Hematological disorders | 2 | | | 2 |
| | | 3 | | | 1 |
| Ribociclib | Hematological disorders | 3 | | | 3 |

**Table 4.** *Cont.*

| Medicines under AM | Type of the 1st Suspected Toxicity | Grade of the 1st Suspected Toxicity | Type of the 2nd Suspected Toxicity | Grade of the 2nd Suspected Toxicity | Number of Patients |
|---|---|---|---|---|---|
| Trametinib + Dabrafenib | Fever | 1 | | | 1 |
| Vandetanib | QT prolongation | 1 | Skin disorders | 2 | 1 |
| Venetoclax | Hematological disorders | 1 | | | 1 |
| Trifluridine + tipiracil | Gastrointestinal disorders | 3 | | | 1 |
| | Hematological disorders | 1 | | | 1 |
| | | 2 | | | 3 |
| | Proteinuria | 2 | | | 1 |
| N total | | | | | 29 |

A total of 33 (63.5%) of the 52 patients included in the study discontinued their treatment. The causes of discontinuation included four reasons, namely: the therapeutic scheme was completed, hyperbilirubinemia, disease progression (DP), and other types of toxicity. DP was the main discontinuation cause, involving 26 patients. In this context, all patients taking the most prescribed drug in the study, trifluridine plus tipiracil, stopped their treatment. The same was verified in patients taking cabozantinib, entrectinib, and ixazomib.

Only four patients that were taking lorlatinib, alectinib, vandetanib, and lenalidomide interrupted their treatment due to toxicity. In this context, one of the two patients that were taking lorlatinib interrupted the treatment due to hyperbilirubinemia development. The only patient taking vandetanib for the treatment of medullary thyroid carcinoma discontinued treatment due to toxicity, namely QT prolongation. Table 5 describes the discontinuation causes for the drugs involved in the study.

**Table 5.** Causes of discontinuation according to the prescribed drug.

| Causes of Drug Discontinuation | Medicines under AM | Number of Patients |
|---|---|---|
| Complete therapeutic scheme | Ixazomib | 1 |
| | Lenalidomide | 2 |
| Hyperbilirubinemia | Lorlatinib | 1 |
| Disease progression | Trifluridine + tipiracil | 5 |
| | Alectinib | 3 |
| | Entrectinib | 1 |
| | Osimertinib | 4 |
| | Trifluridine + tipiracil | 2 |
| | Trifluridine + tipiracil | 2 |
| | Palbociclib | 1 |
| | Ribociclib | 1 |
| | Alectinib | 1 |
| | Niraparib | 1 |
| | Cabozantinib | 1 |
| | Ixazomib | 2 |
| | Lenalidomide | 2 |
| Toxicity | Alectinib | 1 |
| | Vandetanib | 1 |
| | Lenalidomide | 1 |
| N total | | 33 |

The treatment response was analyzed using RECIST 1.1 criteria every 3 months during the study period. After 3 months of treatment, out of a total of 38 patients, 34 patients did not reveal DP. Most of the patients had a partial response (PR) to the treatment, followed by stable disease (SD). Only four patients had DP. After 12 months of treatment, nine patients did not reveal DP. However, in terms of proportion, the percentage of patients with DP was very similar to that of the patients with no DP. These results are described in Table 6.

**Table 6.** Overall treatment response in patients with measurable disease.

| DP | Type of Response | 3 M Evaluation Number of Patients | 12 M Evaluation Number of Patients |
|---|---|---|---|
| No DP | CR | 3 (7.9%) | 4 (25.0%) |
| | PR | 19 (50.0%) | 3 (18.8%) |
| | SD | 12 (31.6%) | 2 (12.5%) |
| DP | | 4 (10.5%) | 7 (43.75%) |
| N total | | 38 | 16 |

CR, complete response; PR, partial response; SD, stable disease; DP, disease progression; M, months.

## 4. Discussion

This study intended to investigate, through an AP model, data on the safety and effectiveness of medicines under AM that were prescribed and dispensed in the LHUM from 2019 to 2021.

The first phase of the study included the implementation of risk minimization measures. These measures aimed to share knowledge with patients and healthcare professionals, particularly to clarify the AM concept, the meaning of the black inverted triangle, and the most frequent AEs reported in the SmPC of the drugs included in the study. The collaboration of all, especially the input from patients and healthcare professionals, is essential for effective pharmacovigilance and benefit–risk management [24,25]. After the prescription of medicines under AM, the AEs reported in medical and pharmaceutical consultations were monitored. These data and other relevant information, such as the causes of drug discontinuation and treatment responses, were analyzed. The main results are discussed below.

A total of 52 patients were included in the study; 31 were female and 21 were male. Most patients were between 60 and 80 years old. The most prescribed and dispensed medicine under AM was trifluridine plus tipiracil. This drug association is indicated for the treatment of adult patients with metastatic colorectal and gastric cancer [26]. In total, eight patients were treated with ixazomib and lenalidomide for multiple myeloma [27,28]. For hepatocellular carcinoma and medullary thyroid carcinoma, only one patient for each was enrolled in the study, taking cabozantinib and vandetanib, respectively [29,30]. Cabozantinib is also indicated in renal cell carcinoma and differentiated thyroid carcinoma [29].

Only 29 patients developed at least one type of toxicity to the drugs studied. Of these, 13 patients developed hematological disorders with a toxicity grade ranging between 1 and 3. Although these drugs have differences in the mechanisms of action, hematological disorders were the most reported ADRs in the study. These AEs occurred particularly in patients taking ixazomib, lenalidomide, palbociclib, ribociclib, venetoclax, or trifluridine plus tipiracil. However, only patients taking ixazomib, palbociclib, or ribociclib developed grade 3 toxicity involving hematological disorders. According to data described in the SmPC, neutropenia, leukopenia, anemia, and thrombocytopenia are considered very common AEs in patients taking palbociclib and ribociclib [31,32]. Based on a pooled dataset from 3 randomized studies, grade 3 neutropenia was developed by 500 (57.3%) patients out of a total of 872 taking palbociclib [31]. For patients taking ribociclib, based on phase III studies results, grade 3 or 4 was reported in 62% of patients [32]. These results may be explained by similarities in the mechanism of action. Palbociclib and ribociclib are cyclin-dependent kinase (CDK) 4/6 inhibitors and are indicated as a treatment option for patients with HR+,

HER2- advanced breast cancer, either as a first-line therapy combined with an aromatase inhibitor or as second-line therapy in combination with fulvestrant [33].

Additionally, four patients reported having more than one toxicity. These patients were taking alectinib, cabozantinib, entrectinib, or vandetanib. In this context, patients that were taking alectinib reported several toxicities, such as gastrointestinal, liver, lung, and kidney disorders. In fact, one of the patients had severe hepatotoxicity with alectinib, similar to what was reported in another real-world study, where hepatic disorders were considered a common ADR with significant identified risk (all grades, 19.8%; grade $\geq$ 3, 2.0%) [34]. The SmPC describes hepatobiliary disorders as common AEs, namely increased aspartate aminotransferase (AST), increased alanine aminotransferase (ALT), and increased bilirubin [35]. According to the literature, drug-induced hepatotoxicity is more common with crizotinib or ceritinib than alectinib. Approximately 1–5% of patients experienced alectinib-induced liver damage [36]. A case report described a patient diagnosed with advanced lung adenocarcinoma treated with alectinib as a first-line therapy. After the initiation of alectinib, the tumor decreased rapidly. On day 79 of treatment, the serum levels of AST and ALT increased to grade 3, according to the CTCAE criteria. Alectinib was immediately discontinued [37]. In our study, one patient discontinued the treatment due to toxicity. However, for this patient in particular, who developed severe hepatotoxicity, the cause of discontinuation was DP. The patient that was taking vandetanib had treatment discontinuation due to toxicity. QT prolongation is a very common AE in patients taking this drug [30]. In our study, QT prolongation might have been the cause of drug discontinuation. Electrocardiography and serum levels of calcium, potassium, and magnesium should be obtained at baseline and during weeks 2 to 4, weeks 8 to 12, and every 3 months thereafter during the therapy [38].

In addition, the patient taking cabozantinib also developed two types of toxicity, mucositis grade 3 toxicity and hypertension grade 4 toxicity. Oral mucositis with cabozantinib was reported at a frequency of 36% in the CABOSUN trial, and a few patients (5.1%) experienced grade 3 or 4 oral mucositis [39]. Likewise, a total of 81% of patients in the CABOSUN trial reported hypertension, with grade 3/4 hypertension having an incidence of 28% [40]. According to the guidelines, after cabozantinib initiation, blood pressure should be monitored early and regularly, and appropriate antihypertensive therapy should be considered if needed. Cabozantinib should be discontinued if hypertension is severe and persistent despite anti-hypertensive therapy and dose-reduction implementations [29]. Nevertheless, in our study, the cause of cabozantinib discontinuation was DP.

All the patients taking lenalidomide had treatment discontinuation. In a total of five patients that were taking this medicine, two patients interrupted the drug due to the completion of the treatment, and two patients discontinued due to DP. Only one patient interrupted the treatment due to toxicity. Neurological disorders were the suspected ADR that might have caused drug discontinuation. Patients with new or worsening neurological symptoms should be monitored [28]. In the literature, although rare, cases of progressive multifocal leukoencephalopathy have been reported [38,40,41].

Lorlatinib was reported to cause dyslipidemia in one of the two patients treated with this drug. A study of patients (N = 295) treated with lorlatinib at 100 mg once daily revealed hypercholesteremia of any grade in 243 patients [42]. However, this AE was not considered the main cause of treatment discontinuation. In our study, hyperbilirubinemia was the toxicity that led to drug discontinuation. Zhu et al. described a real-world data analysis for the efficacy and safety of lorlatinib. Of a total of 95 patients, only 1 reported an increase in blood bilirubin [43]. Lorlatinib is metabolized by the liver. Any hepatic impairment is likely to increase blood bilirubin concentration [44].

In fact, only four patients had treatment discontinuation due to toxicity, which may demonstrate the safety profile of these drugs. The main cause of drug discontinuation was DP. This result was also highlighted through RECIST 1.1 criteria. After 12 months of treatment, the overall number of patients with measurable disease that had DP (43.75%)

was almost the same as the patients that had CR, PR, or SD (56.25%). However, after 3 months of treatment, the overall results were more encouraging.

For this study, some strengths and limitations must be considered. The major strength was the data collected on the safety and effectiveness of drugs that have very few studies described in the literature, particularly in the real-world setting. It is important to note that, for medicines under additional monitoring, even the absence of adverse reactions not described in clinical trials or the SmPC is extremely important postmarketing information for regulatory authorities and healthcare professionals. These real-world data are increasingly being used to evaluate the safety of innovative antineoplastic therapies, such as tumor-targeted therapy. This is particularly useful for assessing drug toxicity profiles in patient populations that are often excluded from randomized clinical trials, such as older patients (such as those included in the present study) who are often patients with comorbidities or poor performance statuses. Furthermore, these antineoplastic treatments can represent a challenge in daily clinical practice, especially in critical and frail subpopulations (such as older patients or polymedicated patients) or in complex socio-health conditions (such as those determined by the recent COVID-19 pandemic that affected the patients included in the present study). We were able to analyze the suspected ADRs and tumor growth in a small, heterogeneous population, which are usually not considered in premarketing clinical trials. In addition, a commitment to pharmacovigilance purposes is notorious among the healthcare professionals and patients who participated in the study.

However, some limitations of our study should also be mentioned. The patient sample size was relatively small, since data were collected from 52 patients only, and, in addition, from a single hospital institution. In this way, the results of our case series study cannot be generalized, but must be analyzed together with other similar studies, carried out in other hospital institutions, either in the same country or in other countries and involving other oncologic patients. The follow-up period of treatment was limited to 12 months, which may not be long enough to fully evaluate the safety and effectiveness of the drugs studied. Furthermore, considering the study period, due to the COVID-19 pandemic, some consultations were delayed or, in some cases, did not happen. In this context, data collection, namely the ADR report and overall response, may have been influenced. Similarly, the study did not assess the impact of drug interactions or other factors (e.g., drug-food interactions) that may contribute to ADRs, nor did it control for confounding factors, such as patient characteristics or comorbidities, which may also have an impact on the risk of ADRs.

Despite these limitations, this is an original study with real-world data that may be eligible for inclusion in a systematic review, designated to better determine the frequency of ADRs, as well as the frequency of their severity grades, obtained in the real-life setting and for certain subgroups of patients (e.g., according to age, sex, etc.) with oral antineoplastic drugs under AM.

## 5. Conclusions

The application of the developed AP model to oral antineoplastic agents under AM has contributed to better management of toxicity and, therefore, to obtain better real-world clinical outcomes for patients. Risk minimization measures were implemented. Patient engagement was crucial to monitor the toxicity in a timely manner. Patients felt better supported, more confident in the treatment instituted, and more encouraged to notify of AEs. The safer use of drugs was promoted, and the quality of services provided by hospital pharmacists was improved. The collaboration and communication between healthcare professionals of the various services involved, enabling teamwork in a multidisciplinary context, are also to be underlined.

We believe that further high-quality clinical studies should be conducted on drugs labeled with the black inverted triangle. This model may be applied and potentially extended to other classes of medicines under AM.

**Supplementary Materials:** The following supporting information can be downloaded at: https://www.mdpi.com/article/10.3390/curroncol30040315/s1, Table S1: Chemical structures and suspected toxicities of the studied oral antineoplastics.

**Author Contributions:** Conceptualization, S.P.C.d.S., M.J. and M.M.; methodology, S.P.C.d.S., M.J. and R.C.e.S.; validation, S.P.C.d.S., M.J., R.C.e.S. and M.M.; formal analysis, S.P.C.d.S., M.J., R.C.e.S., F.R., M.T.H., A.P.D. and M.M.; investigation, S.P.C.d.S., M.J., R.C.e.S. and M.M.; writing—original draft preparation, S.P.C.d.S. and M.J.; writing—review and editing, F.R., M.T.H., R.C.e.S., A.P.D. and M.M.; supervision, A.P.D. and M.M.; project administration, S.P.C.d.S., A.P.D. and M.M. All authors have read and agreed to the published version of the manuscript.

**Funding:** This work was partially supported by CICS-UBI, which is financed by national funds from Fundação para a Ciência e a Tecnologia (FCT) and by Fundo Europeu de Desenvolvimento Regional (FEDER) under the scope of PORTUGAL 2020 and Programa Operacional do Centro (CENTRO 2020), with the project reference numbers UIDB/00709/2020 and UIDP/00709/2020.

**Institutional Review Board Statement:** The study was conducted in accordance with the Declaration of Helsinki and approved by the Ethics Committee of the Local Health Unit of Matosinhos (LHUM) (Porto, Portugal) (protocol code NS 64/CE/JAS-07/06/2019, approval date of 7 June 2019).

**Informed Consent Statement:** Informed consent was obtained from all subjects involved in the study.

**Data Availability Statement:** Data supporting this study consist of irreversibly anonymized clinical data and are included within the article.

**Conflicts of Interest:** The authors declare no conflict of interest.

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
