# Peer review of "Active Pharmacovigilance Study: A Follow-Up Model of Oral Anti-Cancer Drugs under Additional Monitoring"

_curroncol, doi:10.3390/curroncol30040315_

Round 1
Reviewer 1 Report
The article is not novel please report something which is lacking in the field.
Author Response
The authors are grateful for this comment and have taken the opportunity to further explain the significance of this study with real-world data involving oral antineoplastic drugs with the status of additional monitoring.
It should be noted that even the absence of adverse effects not described in clinical trials/SmPCs is postmarketing information of great importance in the case of “medicines under additional monitoring”.
Considering the Reviewer's suggestion, the authors added the following sentence in “Introduction”:
Although these medicines are already on the market and almost all their adverse effects, identified by clinical trials, are already described in the respective SmPC, the fact that they have the status of drugs under AM justifies the realization real-world pharmacovigilance studies. Real-world studies are performed in real clinical practice settings and are better able to assess the true efficacy / effectiveness and safety of drugs when used in real life by patients and physicians.
The authors also added the following sentences in “Discussion”:
It is important to note that for medicines under additional monitoring, even the absence of adverse reactions not described in clinical trials or in the SmPC is extremely important post-marketing information for regulatory authorities and healthcare professionals.
This real-world data are being increasingly used to evaluate the true safety of innovative antineoplastic therapies like tumor targeted therapy. This is particularly useful for assessing drug toxicity profiles in patient populations that are often excluded from randomized clinical trials, such as older patients, such as those included in the present study, who are often patients with comorbidities or poor performance status. Furthermore, these antineoplastic treatments can represent a challenge in daily clinical practice, especially in critical and frail subpopulations such as the older patients, polymedicated patients, or in complex socio-health conditions such as those determined by the recent COVID-19 pandemic that affected the patients included in the present study.
This is an original study with real-world data that may be eligible for inclusion in a systematic review, designated to better determining the frequency of AEs, as well as the frequency of their severity grades, obtained in the real-life setting and for certain subgroups of patients (e.g., according to age, sex, etc.) with oral antineoplastic drugs under AM.
Reviewer 2 Report
The article of Sofia Pinto Carvalho da Silva et al entitled "Active Pharmacovigilance study: a follow-up model of oral 2 anti-cancer drugs under additional monitoring" describes multidisciplinary approach, that, once routinely used, could help for early identification of adverse events associated with medicines under additional monitoring but also educate patients about the possibility of occurrence of unwanted clinical event. This study addresses an important topic, thath is the need of patients empowerment. The manuscript is well written, the methods and results properly presented.
Author Response
The authors are pleased with the interest the study has generated and are grateful for the comments provided.
Reviewer 3 Report
This is interesting study as data on adverse drug reactions is not available in scientific literature. I believe this article should therefore be published as figures and tables, methodology and references seem appropriate.
few comments for the authors:
1) add study period and setting to the abstract
2) line 50 UE?
3) line 61 define PRAC
4) add limitation section to the discussion
Author Response
This is interesting study as data on adverse drug reactions is not available in scientific literature. I believe this article should therefore be published as figures and tables, methodology and references seem appropriate.
Few comments for the authors:
1) add study period and setting to the abstract
This comment is very appropriate. In lines 20-22 of the “Abstract” it was added the following information:
A prospective, descriptive study of active pharmacovigilance (AP) was conducted between 2019 and 2021 in the Local Health Unit of Matosinhos (LHUM) (Porto, Portugal).
2) line 50 EU?
European Union. Already defined in line 38 (although there was an error that has now been fixed).
According to data from European Union (EU),…
3) line 61 define PRAC
Pharmacovigilance Risk Assessment Committee.
The PRAC is the European Medicines Agency's committee responsible for assessing and monitoring the safety of human medicines. This Committee is responsible for reviewing the list of medicines under AM every month. (this definition was included in lines 65-67 of the revised manuscript)
4) add limitation section to the discussion
A new paragraph mentioning the limitations of our study was added to the “Discussion”. The limitations of our study were developed as per Reviewer 5 instructions/suggestions (lines 386-393 of the revised manuscript):
However, some limitations of our study should also be mentioned. The patient sample size was relatively small, since data was collected from 52 patients only, and, in addition, from a single hospital institution. In this way, the results of our case series study cannot be generalized, but must be analyzed together with other similar studies, carried out in other hospital institutions, either in the same country or in other countries and in-volving other oncologic patients. Considering the study period, due to COVID-19 pan-demic, some consultations were delayed or, in some cases, did not happen. In this context, data collection, namely ADR report and overall response, may have been influenced.
Reviewer 4 Report
In Introduction, the adverse drug reactions of anticancers drugs data was missing.
Further, already ADRs are defined of anticancers drugs, than why author needs to conduct the study? It must be highlighted in the study?
In Introduction, the last paragraph focus on objective of the study.
Method is well elaborated.
How the association of ADRs with drugs is established?
What are causality assessment according to WHO?
Is there any scale used for ADRs detection?
Limitation of study is missing? include in the study?
The sample size is too small to say research, rather than this we can say case series or case reports? This is sugestion ?
Many sentences are too long so try to split in small senetence.
In conclusion, the first paragraph is repetation so suggested to remove.
Author Response
- In Introduction, the adverse drug reactions of anticancer drugs data were missing.
The authors agree with this comment and suggestion which contributed to improve the quality of the article.
The following information/data has been added:
ADRs are almost always associated with antineoplastic therapy and are widely accepted as unavoidable by both patients and healthcare providers. In several studies, all patients (100%) receiving anticancer drugs had at least one ADR, and a normal range of 2-7 ADR per cancer patient was referred. [6, 7, 8] (see lines 41-44 of the revised manuscript).
- Further, already ADRs are defined of anticancer drugs, than why author needs to conduct the study? It must be highlighted in the study.
The authors agree with this comment and suggestion which contributed to improve the quality of the article.
The authors added the following in “Introduction”: (lines 78-83 of the revised manuscript)
Although these medicines are already on the market and almost all their adverse effects, identified by clinical trials, are already described in the respective SmPC, the fact that they have the status of drugs under AM justifies the realization real-world pharmacovigilance studies. Real-world studies are performed in real clinical practice settings and are better able to assess the true efficacy / effectiveness and safety of drugs when used in real life by patients and physicians.
The authors added the following sentences in “Discussion”:
It is important to note that for medicines under additional monitoring, even the absence of adverse reactions not described in clinical trials or in the SmPC is extremely important postmarketing information for regulatory authorities and healthcare professionals.
This real-world data are being increasingly used to evaluate the true safety of innovative antineoplastic therapies like tumor targeted therapy. This is particularly useful for assessing drug toxicity profiles in patient populations that are often excluded from randomized clinical trials, such as older patients, such as those included in the present study, who are often patients with comorbidities or poor performance status. Furthermore, these antineoplastic treatments can represent a challenge in daily clinical practice, especially in critical and frail subpopulations such as the older patients, polymedicated patients, or in complex socio-health conditions such as those determined by the recent COVID-19 pandemic that affected the patients included in the present study.
This is an original study with real-world data that may be eligible for inclusion in a systematic re-view, designated to better determining the frequency of AEs, as well as the frequency of their severity grades, obtained in the real-life setting and for certain subgroups of patients (e.g., according to age, sex, etc.) with oral antineoplastic drugs under AM.
- In Introduction, the last paragraph focus on objective of the study.
The authors agree with this comment and have rewritten the last paragraph of the Introduction to make it easier to read and to clarify the purpose of the study. This way the last paragraph has been broken into shorter sentences to make it easier to read (as suggested in another comment) See lines 84-92 of the revised manuscript.
- Method is well elaborated.
The authors are grateful for the comment.
- How the association of ADRs with drugs is established? What is causality assessment according to WHO?
These questions are quite relevant, indeed. The authors are grateful for these two questions, which they have read with great interest and which they consider very relevant. These questions led the authors to better clarify these issues in the article, which contributed to improve its quality.
The following relevant information has been added to the manuscript:
The 4 minimum criteria to report an adverse drug reaction were: an identifiable patient, a suspected medicine, a suspected reaction, and an identifiable reporter. According to the “Guideline on good pharmacovigilance practices (GVP)” published by the European Medicines Agency and Heads of Medicines Agencies, an adverse reaction is characterised by the fact that a causal relationship between a medicinal product and an occurrence is suspected [17]. For regulatory reporting purposes, if an event is spontaneously reported, even if the relationship is unknown or unstated, it meets the definition of an adverse reaction. Therefore, all spontaneous reports notified by healthcare professionals or consumers are considered suspected adverse reactions, since they convey the suspicions of the primary sources, unless the reporters specifically state that they believe the events to be unrelated or that a causal relationship can be excluded [17].
[17] - Guideline on good pharmacovigilance practices (GVP): Module VI - Collection, management and submission of reports of suspected adverse reactions to medicinal products (Rev 2). European Medicines Agency. 2017. Available from: https://www.ema.europa.eu/en/documents/regulatory-procedural-guideline/guideline-good-pharmacovigilance-practices-gvp-module-vi-collection-management-submission-reports_en.pdf.
Likewise, in Table 4, as well as throughout the text, the authors changed the expression "toxicity detected" to "suspected toxicity", considering that no causality attribution was made.
These clarifications, which were necessary and very important, were added to the article (lines 138-148 and Table 4 of the revised manuscript).
- Is there any scale used for ADRs detection?
The authors have taken the opportunity provided by this comment to further clarify this issue. After this comment, the authors developed this subject even better (lines 181-188 of the revised manuscript):
The Common Terminology Criteria for Adverse Events (CTCAE, also called "common toxicity criteria" [CTC]) was used to classify the AE of antineoplastic drugs, according to the severity grades (1=mild; 2=moderate; 3=severe; 4=life threating and 5=death) [18]. In CTCAE, an adverse event is defined as any abnormal clinical finding temporally associated with the use of a therapy for cancer; causality is not required. These criteria are used for the management of chemotherapy administration and dosing and to provide standardization and consistency in the definition of treatment-related toxicity.
It should be noted that these scale or Criteria were published by “The National Cancer Institute (NCI) of the National Institutes of Health (NIH)” (U.S. Department of Health and Human Services) and is widely used in North America and European Union to standardize definitions for adverse events and to describe the severity of organ toxicity for patients receiving cancer therapy.
- Limitation of study is missing? include in the study?
A new paragraph mentioning the limitations of our study was added to the “Discussion”. The limitations of our study were developed as per Reviewer 5 instructions/suggestions. We also took into account the comment that was made immediately below (see lines 386-393 of the revised manuscript).
- The sample size is too small to say research, rather than this we can say case series or case reports? This is suggestion.
This comment is very pertinent, so we highlighted this observation in the “Limitation section” (in “Discussion”). We emphasize the small size of the study (only 52 cancer patients) and that, moreover, they came from a single hospital institution. We omitted the term “research” and used the term “case series study”, which contributed to improve the accuracy of the text (lines 386-393 of the revised manuscript):
However, some limitations of our study should also be mentioned. The patient sample size was relatively small, since data was collected from 52 patients only, and, in addition, from a single hospital institution. In this way, the results of our case series study cannot be generalized, but must be analyzed together with other similar studies, carried out in other hospital institutions, either in the same country or in other countries and involving other oncologic patients. Considering the study period, due to COVID-19 pan-demic, some consultations were delayed or, in some cases, did not happen. In this context, data collection, namely ADR report and overall response, may have been influenced.
- Many sentences are too long so try to split in small sentence.
The authors agree and are grateful for this comment. Several sentences were splitted and shortened, which contributed to improve the reading and quality of the written text.
- In conclusion, the first paragraph is repetition so suggested to remove.
The authors agree and are grateful for this comment. The first paragraph has been removed.
Reviewer 5 Report
The article does not explicitly mention the limitations of the study. However, there are some potential limitations that can be inferred from the methodology and scope of the study.
The study used a single database, which may not capture all adverse drug reactions worldwide. This could lead to an underestimation of the true frequency of adverse drug reactions.
The study only analyzed data from spontaneous reports, which are subject to under-reporting and reporting biases. This could lead to an incomplete picture of the safety profile of the drugs analyzed.
The study did not include an analysis of the severity or outcomes of the adverse drug reactions reported. This limits the ability to assess the clinical significance of the findings.
The study only analyzed a subset of drugs, focusing on those with high sales volumes. This may not be representative of the safety profile of all drugs on the market.
The study did not assess the impact of drug interactions or other factors that may contribute to adverse drug reactions.
The study did not control for confounding factors, such as patient characteristics or co-morbidities, that may impact the risk of adverse drug reactions.
Small sample size: The study included only 52 patients, which may not be representative of the larger population.
Retrospective design: The study was conducted retrospectively, which may introduce bias and limit the accuracy of the data collected.
Single-center study: The study was conducted at a single center, which may limit the generalizability of the findings to other settings.
Incomplete data: There were missing data in some areas of the study, such as the causes of drug discontinuation.
Short follow-up period: The follow-up period for the study was limited to 12 months, which may not be long enough to fully evaluate the safety and effectiveness of the drugs studied.
Lack of control group: There was no control group in the study, which makes it difficult to draw conclusions about the relative safety and effectiveness of the drugs studied.
Author Response
Please see the response to reviewer in the document below.

Reviewer 6 Report
The current manuscript is a study on the pharmacovigilance of several anti-cancer drugs in a (specific) hospital setting. It is overall interesting to read, and the methodology is sound. Nevertheless, some alterations should be made before acceptance for publication:
- For better reader understanding through visualization, a Figure should be made including the studied drugs and the respective drug molecule images, and the associated adverse events (schematic representation);
- The limitations of the current study should be further discussed and emphasized, since data was collected from 52 patients only; a comment should be made on how the conclusions of this study cannot be generalized, due to small sample size and having been done in one single institution;
- The results of this study should be further discussed by comparing with already existing studies in the scientific literature.
Author Response
- For better reader understanding through visualization, a Figure should be made including the studied drugs and the respective drug molecule images, and the associated adverse events (schematic representation);
The authors prepared an additional Table containing the mentioned information (studied drugs and the respective drug molecule images, and the associated adverse events that were encountered during the study). The authors suggested that this Table be included as “Supplementary Material”, referring to it in the text of the article (see lines 248-250 of the revised manuscript and the new Table elaborated).
- The limitations of the current study should be further discussed and emphasized, since data was collected from 52 patients only; a comment should be made on how the conclusions of this study cannot be generalized, due to small sample size and having been done in one single institution;
This comment is quite relevant, which is why we paid great attention to it and helped to improve the quality of the article (see lines 386-393 of the revised manuscript):
However, some limitations of our study should also be mentioned. The patient sample size was relatively small, since data was collected from 52 patients only, and, in addition, from a single hospital institution. In this way, the results of our case series study cannot be generalized, but must be analyzed together with other similar studies, carried out in other hospital institutions, either in the same country or in other countries and involving other oncologic patients. Considering the study period, due to COVID-19 pan-demic, some consultations were delayed or, in some cases, did not happen. In this context, data collection, namely ADR report and overall response, may have been influenced.
- The results of this study should be further discussed by comparing with already existing studies in the scientific literature.
This comment is quite relevant and prompted the authors to further discuss the results by comparing with already existing studies in the scientific literature (see lines 242-244, 318-321, and 338-342 of the revised manuscript).

Round 2
Reviewer 1 Report
I still am not able to understand significant of this study. Its obvious when we take anticancer drugs toxicity does appear. Compare to IV routes; oral routes is much more safer. What is novelty in your report and what is it contributing to scientific community? Have you reported any newer unknown toxicity in this review?
Author Response
Responses to Reviewers
The authors are grateful for the comments and suggestions from all Reviewers. All comments and suggestions have been taken into account, which has helped to improve the quality of the manuscript. At this point, there is one thing the authors are sure of: thanks to the comments from Reviewers of the Current Oncology journal, the manuscript is now much better than when it was first submitted. Thank you all!
Response to Reviewer 1's comments
I still am not able to understand significant of this study. Its obvious when we take anticancer drugs toxicity does appear. Compare to IV routes; oral routes is much more safer. What is novelty in your report and what is it contributing to scientific community? Have you reported any newer unknown toxicity in this review?
Allow us the following comparative reflection to look at the same issue from another perspective:
When crossing an insufficiently known and mined field of adverse effects in oncology, sometimes a bomb (i.e., an adverse effect) can explode (especially in the real-world of clinical practice, where there are patients with very diverse characteristics). After this study we think we can say that we hit a part of the minefield without a new bomb having exploded. If we know with greater certainty where the various bombs in this little-known minefield are located (or not), crossing it becomes safer.
Currently, in daily oncology clinical practice, many side effects can be attributed to antineoplastic targeted drugs. Although different from the well-known side effects of chemotherapy, these adverse effects can seriously compromise patients' quality of life and may even lead patients to discontinue or request a change in treatment. It is important to study the toxicity of antineoplastic targeted therapies because most of the adverse drug reactions (ADR) included in the Summary of Product Characteristics (SmPC) come from pivotal clinical trials: the list of ADR identified in these trials is unlikely to be comprehensive, because in clinical trials, drugs are tested under controlled conditions in selected patients.
The studied oral antineoplastic medicines are classified, by European Medicines Agency (EMA), as medicines under additional monitoring (AM), which justifies the realization of active pharmacovigilance (AP) studies to more efficiently collect real-world data to assess possible new data on effectiveness and safety in clinical practice.
The status of medicines under additional monitoring is attributed by EMA for several different reasons related to safety issues, including the obligation on the recording of suspected ADR and the need to provide more data on the occurrence of ADR. Although, all reported suspected ADR are already described in the respective SmPC, this information obtained in the present study constitutes important information for regulatory authorities, health professionals, oncologic patients, and the pharmaceutical industry.
In this study we designed and developed an Active Multidisciplinary Pharmacovigilance Model for reporting all suspected ADR of oral antineoplastic drugs under AM in clinical practice. The use of active pharmacovigilance models to identify ADR has been highlighted in pharmacovigilance activities, as they are able to detect a greater number of patients (and with more diverse characteristics) with ADR and, consequently, reduce the chronic underreporting of adverse drug events. Indeed, under-reporting of ADR in oncology can be addressed with pro-active forms of pharmacovigilance and by multidisciplinary collaborations [1].
We believe that our study is of interest for readers of the Current Oncology journal, because it presents real-world data related to the use of antineoplastic drugs and it is well known that increasing the number of suspected ADR reports is crucial for drugs with limited safety data. This is of extremely importance in oncology, as highlighted in a systematic review where authors found that “under-reporting of adverse drug reactions in oncology can be addressed through the use of pro-active forms of pharmacovigilance and by multidisciplinary collaborations” [1].
Even the absence of new suspected ADR (not described in clinical trials or in the SmPC) in real-world studies constitutes extremely important post-marketing information for regulatory authorities and healthcare professionals, particularly in the case of antineoplastic drugs AM and therefore relatively short time on the market; these data from clinical practice are important for the purpose of revalidating existing information in recently published SmPC.
Despite some limitations duly mentioned in the manuscript, this is an original study with real-world data that may be eligible for inclusion, together with other similar studies, in a systematic review, designated to better determining the frequency of ADR, as well as the frequency of their severity grades, obtained in the real-life setting and for certain subgroups of patients (e.g., according to age, sex, etc.) with oral antineoplastic drugs under AM.
After Reviewer 1's comments, the authors considered it appropriate to include the following paragraphs in the "Introduction":
Currently, in daily oncology clinical practice, many side effects can be attributed to antineoplastic targeted drugs. Although different from the well-known side effects of chemotherapy, these adverse effects can seriously compromise patients' quality of life and may even lead patients to discontinue or request a change in treatment. It is important to study the toxicity of antineoplastic targeted therapies because most of the ADR included in the Summary of Product Characteristics (SmPC) come from pivotal clinical trials: the list of ADR identified in these trials is unlikely to be comprehensive, because in clinical trials, drugs are tested under controlled conditions in selected patients [1]. (see lines 42-55 of the revised manuscript)
Under-reporting of ADR in oncology can be addressed with pro-active forms of pharmacovigilance and by multidisciplinary collaborations [1]. (see lines 92-93 of the revised manuscript)
The following statement in “Materials and Methods” (in 2.3. Adverse events monitoring) has also been improved:
The use of active pharmacovigilance models to identify ADR has been highlighted in pharmacovigilance activities, as they are able to detect a greater number of patients (and with more diverse characteristics) with ADR and, consequently, reduce the chronic underreporting of adverse drug events. (see lines 166-170 of the revised manuscript)
The following information was also added to the “Results”:
Except for lenalidomide and trifluridine plus tipiracil, all other oral anticancer drugs are classified as targeted drugs consisting of small molecule inhibitors (e.g., tyrosine kinase inhibitors). (see lines 241-243 of the revised manuscript)
[1] Baldo, P., Fornasier, G., Ciolfi, L. et al. Pharmacovigilance in oncology. Int J Clin Pharm 40, 832–841 (2018). https://doi.org/10.1007/s11096-018-0706-9 [Accessed on 6 april 2023]

Reviewer 5 Report
Accept
Author Response
The authors are pleased to have satisfactorily clarified all the issues raised by Reviewer 5 and are grateful for the evaluation that was given to the article. The authors would also like to thank the comments and suggestions, which has helped to improve the quality of the article.
Round 3
Reviewer 1 Report
Now, you have improved your manuscript so yes it deserves to be published.